# Cross-cultural adaptation, reliability and validity of the Turkish version of the Smart Tools Proneness Questionnaire (STP-Q)

Serdar Yılmaz Esen[1], Ceyhun Türkmen[2]*, Tülin Düger[3]

1 Graduate School of Health Sciences, Hacettepe University, Ankara, Turkiye, 2 Department of Occupational Therapy, Çankırı Karatekin University, Çankırı, Turkiye, 3 Faculty of Physical Therapy and Rehabilitation, Hacettepe University, Ankara, Turkiye

* fztceyhunturkmen@gmail.com

**Data Availability Statement:** Data cannot be shared publicly due to institutional privacy policies. Data access requests can be directed to the Çankırı Karatekin University Ethics Committee. Please

## Abstract

This study elucidates the cross-cultural adaptation, reliability, and validity of the Turkish version of the Smart Tools Proneness Questionnaire (STP-Q), designed to probe into individuals' engagement with smart tools within the Turkish cultural milieu. Undertaking a rigorous adaptation process, this investigation aimed to ensure the questionnaire's relevance and intelligibility, subsequently, assessing its psychometric properties within a cohort of 387 participants. The exploratory factor analysis revealed a tripartite structure that reflects the original instrument, covering utilitarian use, hedonic and social use, and the inclination to delegate tasks. This congruity underscores the STP-Q's adeptness in capturing the complex dimensions of smart tool interaction across various contexts. Demonstrated by a Cronbach's alpha of 0.954 and a test-retest reliability index of 0.851, the results affirm the questionnaire's exceptional internal consistency and significant temporal stability. Further, the execution of convergent validity assessments alongside the E-Learning Readiness Scale and the Nomophobia Questionnaire augmented the STP-Q's validity, unveiling correlations that delineate the intricate interrelations among smart tool proneness, e-learning readiness, and nomophobia. Conclusively, the STP-Q distinguishes itself as a reliable and valid instrument for gauging tendencies towards smart tool use among the Turkish populace, providing profound insights into digital behavior across different cultural backgrounds. Its confirmed three-factor structure and robust psychometric attributes render it an indispensable resource for both individual assessments and expansive digital behavior investigations, enabling cross-cultural comparisons and enhancing our understanding of technology engagement dynamics.

## Introduction

The rise of smart tools has significantly impacted, our daily lives, transforming communication, productivity, and leisure activities [1]. "These tools have automated tasks and provided up-to-date information, enhancing everyday convenience. However, they also raise concerns

contact [Assoc. Prof. Dr. İlknur Göl (Vice President of Ethics Committee) at [ilknur@karatekin.edu.tr] for researchers who meet the criteria for access to confidential data."

**Funding:** The author(s) received no specific funding for this work.

**Competing interests:** The authors have declared that no competing interests exist.

about privacy and security [2]. It is crucial for individuals and society to find a balance between the benefits and drawbacks of these technologies. In addition to smart tools, digital technologies have transformed how people interact with their surroundings and each other [3]. For example, voice-activated assistants and surveillance cameras have simplified remote household management and improved security protocols [4]. However, these devices also expose vulnerabilities for hackers, who can access them and compromise personal data. To fully benefit from smart home technology, users must strike a balance between convenience and privacy [5]. Hackers have successfully manipulated voice assistants and security cameras, leading to intrusive surveillance and potential data breaches. For example, private chats of voice assistants could be recorded and transmitted to an unknown individual without their consent. While smart home tools offer convenience, they also pose significant privacy and security risks if not adequately safeguarded [6].

The Smart Tools Proneness Questionnaire (STP-Q) is a significant instrument for evaluating individuals' inclination to utilize smart tools, which are becoming more prevalent in the modern digital era. STP-Q was first created by Navarro et al. to evaluate individuals' inclination to utilize smart tools [7]. The questionnaire underwent a meticulous development process that involved creating items based on a comprehensive literature review and conducting initial validation experiments with diverse samples. The initial validation process showed that the questionnaire had strong psychometric properties, including high internal consistency and test-retest reliability. The STP-Q has subsequently been employed in numerous studies to examine the correlation between smart tool utilization and other psychological and behavioral consequences, underscoring its significance in comprehending technology adoption trends [8, 9]. It aids businesses in identifying potential obstacles or difficulties that individuals may have when adopting and utilizing smart solutions, enabling them to develop targeted plans for a more seamless transition for their employees. The STP-Q is a crucial tool for enhancing comprehension of technology adoption and usage patterns across different demographic segments.

"The Smart Tools Proneness Questionnaire (STP-Q) provides a thorough evaluation of individuals' inclination to utilize smart tools. It differentiates itself from other similar measures like the Technology Acceptance Model (TAM) [10], the Mobile Phone Problem Use Scale (MPPUS) [11], and the Internet Addiction Test (IAT) [12]. Although TAM largely emphasizes perceived usefulness and ease of use as factors that determine technology acceptance, it does not explore the intricate behaviors and attitudes towards smart technologies that are measured by the STP-Q. MPPUS focuses on addressing the issue of problematic mobile phone use and its psychological causes, providing valuable insights into certain areas of addiction to mobile technology. However, it falls short in providing a broader view of overall smart tool usage, which is offered by the STP-Q. Similarly, the IAT assesses the extent of internet use that disrupts daily functioning, but it does not include the utilitarian, hedonic, and social aspects of smart tool use that are addressed by the STP-Q. The STP-Q's incorporation of several dimensions makes it well-suited for the Turkish setting, where cultural, social, and economic factors may influence technology adoption in various ways. By considering a wider range of behaviors, the STP-Q offers a comprehensive understanding of smart tool usage, making it an excellent instrument for researchers and practitioners who aim to study and address technology use within the Turkish community".

The adaptation of psychometric instruments to different cultural contexts is of paramount importance, as it ensures that the tools accurately reflect the specific cultural nuances that influence behavior and attitudes towards technology. This is particularly relevant for the Smart Tools Proneness Questionnaire (STP-Q) as technology usage and attitudes can vary significantly across cultures due to differences in technological infrastructure, digital literacy, and societal norms. Studies such as those by Lee et al. have demonstrated significant variations in

the adoption and use of technology in United States versus South Korea contexts, influenced by cultural values and communication styles [13]. Therefore, adapting the STP-Q for the Turkish population not only addresses linguistic differences but also cultural factors that could impact the interpretation and relevance of its items. By ensuring that the questionnaire resonates with the cultural realities of the Turkish context, the adapted STP-Q can provide more accurate and meaningful insights into the predisposition towards smart tools, aiding in the development of culturally appropriate technologies and interventions that are more likely to be adopted and effective.

In the field of cross-cultural work, the need for culturally sensitive and linguistically precise assessment tools is increasing. The propensity to use smart technologies may vary across cultures as a result of inequalities in technological availability, cultural standards, and social principles. Accurate and reliable translations of such tools ensure that these cultural subtleties are successfully captured [14]. Understanding global patterns in technology adoption and use is crucial for researchers. This study aims to increase the generalizability of findings in the rapidly growing field of smart tools use by modifying the STP-Q for the Turkish population. Additionally, it aims to bridge cultures and contribute to a more comprehensive and inclusive understanding of this topic. The adaptation and validation of the STP-Q to Turkish culture would enhance our comprehension of smart tools utilization in Turkiye by furnishing insights into technology adoption trends and preferences within the local community.

## Materials and methods

### Participants

This methodological and cross-sectional study translates, the STP-Q into Turkish and aims to reveal its reliability and validity. The individuals participating in the research were selected from all regions of Turkiye. All participants must meet the following inclusion criteria: (i) being over 18 years of age, (ii) using smart tools and equipment (iii) provide written consent to participate. All participants will sign a consent form explaining the purposes and procedures of this study. Informed consent will be obtained from all individual participants included in the study. Participants with certain abnormalities, such as cognitive or neurological deficits, pain in the upper extremities, functional limitations, or cognitive impairments, will be excluded from this study to establish accurate norms. For the sample size, 10 people are needed for each item, for a total of at least 270 people [15]. The recruitment period for this study started on December 15, 2023, and ended on February 5, 2024. For the translation of the Turkish version of the STP-Q questionnaire, permission was obtained from the developer of the STP-Q questionnaire.

### Ethics statement

The study received ethical approval from the Çankırı Karatekin University Ethics Commission (decision code: ab2e5654208645eb) on February 6, 2023. All procedures conducted in this research involving human participants adhered to the ethical standards established by the institutional and national research committees, in compliance with the Declaration of Helsinki and its subsequent amendments or analogous ethical guidelines. Informed written consent was obtained from all participants included in the study. Participants were thoroughly informed about the study's objectives, methodologies, potential risks, and benefits. They were also provided with sufficient time to consider their involvement and to ask any pertinent questions.

## Translation and cross-cultural adaptation

The translation of the STP-Q from English to Turkish was conducted following the guidelines established by Beaton et al., Terwee et al., and Heim et al., ensuring a rigorous cross-cultural adaptation and validation process for self-report measures [15–17].

The translation process of the STP-Q into Turkish adhered to a methodical and well-organized process to guarantee linguistic and cultural precision. The criteria for selecting translators included native-level fluency in both English and Turkish, along with professional expertise in translating academic and psychometric instruments. Two proficient bilingual professionals conducted separate translations of the original English version of the STP-Q into Turkish. Subsequently, a panel of specialists evaluated the translations and rectified any inconsistencies through deliberation and agreement.

To enhance the precision and comparability of the translation, a back-translation was conducted by two separate native English speakers who were not part of the initial translation procedure. The back-translated version was then compared to the original English version to detect any errors or semantic disparities. The translation committee resolved any inconsistencies to produce a final Turkish version that preserved the original instrument's content and purpose.

Before the main study, a preliminary test was conducted with a limited number of Turkish volunteers to assess the clarity and understandability of the translated items. Feedback received from the participants of the pilot test was utilized to make slight modifications to the wording and phrasing of certain elements to improve their clarity. This iterative procedure guaranteed that the final Turkish version of the STP-Q was culturally and linguistically suitable for the intended demographic.

## Instruments

**Smart Tools Proneness Questionnaire (STP-Q).** The STP-Q is a self-reported measure of an individual's propensity to use smart tools. STP-Q consists of 27 items and is rated on a 7-point Likert-type scale (From 1: completely disagree to 7: completely agree). STP-Q offers practitioners a measure of individual propensity to use smart tools along three dimensions: utilitarian use, hedonic and social use, and proneness to task delegation [7].

**E-learning readiness scale.** The original scale was developed to measure pre-service teachers' e-learning readiness levels. The scale consists of five subscales: self-competence, self-directed learning, motivation, perceived usefulness, and financial competence. The scale consists of 17 items and is rated on a 5-point Likert-type scale (from 1: strongly disagree to 5: strongly agree) [18, 19].

**Nomophobia Questionnaire (NMP-Q).** Nomophobia is considered a phobia of the modern age that has entered our lives as a by-product of the interaction between people and mobile information and communication technologies, especially smartphones. This questionnaire helps define and explain the dimensions of nomophobia and measures nomophobia. This scale consists of 20 items and a 7-point Likert scale type (from 1: completely disagree to 7: completely agree). The scale consists of four sub-dimensions. It is defined as not being able to communicate, losing connectedness, not being able to access information and giving up convenience [20, 21].

## Statistical analysis

The statistical software IBM SPSS Statistics for Windows version 23.0 developed by IBM Corp. In Armonk, NY, USA, were utilized for statistical studies. Statistical methods such as probability plots and histograms that involve analysis and visualization. A study of continuous data

was conducted using the Kolmogorov-Smirnov and Shapiro-Wilk tests to assess normal distribution. The descriptive analyses presented categorical variables as numerical values and percentages, while continuous variables were presented as the mean, standard deviation, and median (interquartile range). The current study assessed the internal consistency, test-retest reliability, content, construct, and criterion validity of the STP-Q, with a p-value of 0.05 indicating statistical significance. Internal consistency between the items was assessed by Cronbach's alpha coefficient. For comparisons, an alpha of at least 0.70 was "sufficient," an alpha of 0.80 or higher was "good," and an alpha of 0.90 was "excellent" [22]. For test-retest reliability, the intraclass correlation coefficient (ICC) was used. ICC less than 0.50 were considered "poor," between 0.50 and 0.75 "moderate", between 0.75 and 0.90 "good" and > 0.90 "excellent" reliability [23]. The intercorrelation of variables was performed using Spearman's rho correlation coefficient. The results were assessed at a significance level of p<0.05 with a 95% confidence interval.

## Validity

**Content validity.** The participants were queried about the accuracy of the STP-Q in evaluating their inclination towards smart tools. "Are the items on this scale a precise measure of your relationship with smart tools?" Do you believe that the goal of this scale is to assess the extent of inclination towards smart tools? The content was deemed valid if the proportion of positive answers exceeded 90% [24].

**Structure validity.** Exploratory factor analysis (EFA) were used to evaluate the internal construct validity of the STP-Q. The Kaiser Meyer Olkin (KMO) Test and Bartlett's test of sphericity, respectively, were used to determine whether the sample size was adequate and whether the data were appropriate for factor analysis. The factors were retained based on eigenvalues of more than one. To evaluate the factors' goodness of fit, the ratio of the chi-square test of the model fits the degrees of freedom (x2/df) [values of five or less], the Root Mean Square Error of Approximation (RMSEA <0.08 acceptable, <0.05 excellent) andthe Comparative Fit Index (CFI; >0.90 acceptable, >0.95 excellent) were utilized [25].

**Convergent validity.** In order to analyze the external construct validity of the STP-Q, we utilized hypothesis testing to determine its convergent validity. For this analysis, Spearman's correlation coefficients (rho) were used to investigate the anticipated connections between the STP-Q and both the E-Learning Readiness Scale and the NMP-Q.

## Reliability

Internal consistency and test-retest reliability analyzes were examined to determine the reliability of the STP-Q. Internal consistency was assessed by calculating Cronbach's alpha coefficient, while test-retest reliability was assessed by intraclass correlation coefficient (ICC) along with confidence interval and Spearman correlation coefficients. ICC values were interpreted as follows: excellent agreement:>0.90; good agreement: 0.75–0.90; moderate agreement: 0.50–0.75. Correlation coefficient values were interpreted as follows: negligible: 0.00–0.20; weak: 0.21–0.40; medium: 0.41–0.60; strong: 0.61–0.80 and very strong: 0.81–1.00. The t test was used to assess the degree of agreement and systematic variation between test and retest scores, and Bland-Altman determined the 95% agreement threshold.

## Reproducibility

To ensure the accuracy of the measurement technique, Standard Error of Measurement (SEM) and Smallest True Difference (SRD) formulas were used to determine test-retest reliability [26].

**Table 1. Participant demographic characteristics and questionnaire scores.**

| | Mean | Standard Deviation | Median |
|---|---|---|---|
| Age | 27.47 | 9.59 | 22.00 |
| Weight | 66.09 | 13.29 | 65.00 |
| Height | 166.29 | 8.44 | 165.00 |
| BMI | 23.80 | 3.89 | 23.43 |
| STP-Q (total score) | 137.16 | 32.87 | 146.00 |
| STP-Q Factor 1 | 65.51 | 17.56 | 71.00 |
| STP-Q Factor 2 | 40.52 | 11.09 | 43.00 |
| STP-Q Factor 3 | 31.12 | 7.74 | 32.00 |
| E-Learning Readiness Scale | 58.08 | 15.21 | 61.00 |
| NMP-Q | 80.67 | 33.19 | 82.00 |

SD: Standard Deviation, BMI: Body Mass Index, STP-Q: Smart Tools Proneness Questionnaire. NMP-Q: Nomophobia Questionnaire.

## Results

One hundred men (26.7%) and two hundred and seventy-four women (73.3%), totaling 374 individuals with a mean age of 27.47±9.59 years, were included in the present study. The characteristics of the participants are shown in Table 1, with a mean weight of 66.09±13.29 kg, a mean height of 166.29±8.44 cm, and a mean Body Mass Index (BMI) of 23.80±3.89. A thorough examination of demographic data uncovered notable age-related patterns in the utilization of smart tools. Younger individuals exhibited greater levels of smart tool engagement in comparison to older participants (r = -0.450, p = 0.001). This suggests that age plays a significant role. In addition, there was no substantial link observed between BMI and smart tool usage (r = 0.100, p = 0.253), which contradicted our initial premise. The gender analysis revealed that female participants had a higher level of engagement with smart gadgets for social and hedonic goals in comparison to male participants (t(198) = 2.87, p = 0.004).

### Translation, cross cultural adaptation

The translation of the STP-Q items into Turkish was accomplished through the collaborative effort of two bilingual native Turkish speakers, achieving a unanimous agreement without encountering linguistic challenges, notably the absence of idiomatic expressions or complex clauses in the original version. Subsequently, the Turkish version underwent a reverse translation into English by two independent native English speakers, facilitating a comparison with the original questionnaire. This comparative analysis revealed no significant deviations in terms of semantic clarity and grammatical integrity, thereby confirming the linguistic and conceptual fidelity of the Turkish translation. Ultimately, the final English rendition received formal endorsement from the developers of the STP-Q, signifying its acceptability and adherence to the original semantic intent.

### Validity

**Content validity.** The proportion of affirmative responses surpassed the 90% threshold, an indication of robust participant approval. Furthermore, the absence of adverse critiques concerning both the substance and comprehensibility of the questionnaire underscores its acceptability. Consequently, such findings substantiate the instrument's content validity, attesting to its adequacy in capturing the intended constructs.

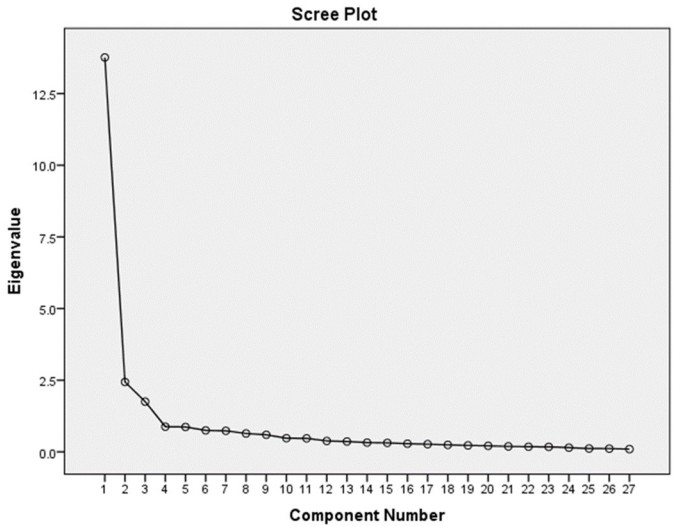

**Fig 1.**

**Structure validity.** The exploratory factor analysis (EFA) conducted on the STP-Q yielded a three-factor solution with eigenvalues surpassing the threshold of one, showcasing the instrument's ability to capture diverse dimensions of smart tool usage (Fig 1). (Fig 1. Scree Plot Illustrating the Factorial Dimensions Derived from Exploratory Factor Analysis of the STP-Q).

Factor 1 emerged as the most significant, with an eigenvalue of 13.76, accounting for substantial variance in the dataset. Predominantly characterized by items 7, 8, 10, 16, 19, 20, 22, and 26, which all register loadings above 0.80, this factor appears to reflect the utilitarian dimension of smart tools use, highlighting users' proficiency and efficiency in employing these technologies.

Factor 2, with an eigenvalue of 1.75, meets the established criteria for retention and is distinctly indicated by a strong loading of 0.726 on item 12. This factor seems to represent the hedonic aspect of smart tool use, suggesting a user's inclination toward the enjoyment and constant connectivity provided by smart technologies.

Factor 3, displaying an eigenvalue of 2.43, is marked by significant loadings, notably a 0.885 on item 4. It delineates a propensity to delegate tasks to smart tools, underscoring a dimension of user interaction that relies on technology to manage responsibilities efficiently (Table 2).

**Convergent validity.** Correlation analysis was employed to ascertain the convergent validity of the STP-Q by examining its associations with the E-Learning Readiness Scale and the NMP-Q. The Spearman's rho correlation coefficient indicated a positive but modest relationship between the STP-Q total score and the E-Learning Readiness Scale (r = 0.110, p = 0.034), suggestive of a slight convergent validity. This implies that while there is a statistically significant correlation between the propensity to use smart tools and e-learning readiness, the overlap between the constructs they measure is relatively limited.

Conversely, the STP-Q total score demonstrated a more pronounced positive correlation with the NMP-Q (r = 0.336, p < 0.001), pointing to a moderate degree of convergent validity. This suggests that individuals' proneness to smart tools has a more substantial and statistically significant relationship with their levels of nomophobia, potentially reflecting an intertwined behavioral pattern between the usage of smart tools and the anxiety experienced in their absence (Table 3).

**Table 2. Factor loadings for the items of the STP-Q.**

| Item | Factor loading | | |
|---|---|---|---|
| | **Factor 1** | **Factor 2** | **Factor 3** |
| 1 | -0.007 | 0.328 | **0.619** |
| 2 | 0.241 | -0.112 | **0.789** |
| 3 | 0.183 | 0.045 | **0.830** |
| 4 | 0.183 | 0.010 | **0.885** |
| 6 | **0.634** | 0.161 | 0.297 |
| 7 | **0.894** | 0.019 | 0.143 |
| 8 | **0.901** | 0.036 | 0.172 |
| 9 | **0.646** | 0.033 | 0.240 |
| 10 | **0.908** | 0.100 | 0.125 |
| 11 | **0.712** | 0.302 | 0.110 |
| 12 | 0.294 | **0.726** | -0.042 |
| 13 | **0.680** | 0.281 | 0.096 |
| 14 | **0.816** | 0.180 | 0.184 |
| 15 | **0.680** | 0.230 | 0.155 |
| 16 | **0.865** | 0.142 | 0.147 |
| 18 | **0.797** | 0.170 | 0.097 |
| 19 | **0.885** | 0.095 | 0.144 |
| 20 | **0.884** | 0.102 | 0.137 |
| 21 | **0.713** | 0.307 | 0.184 |
| 22 | **0.882** | 0.121 | 0.136 |
| 24 | **0.795** | 0.240 | 0.014 |
| 25 | 0.193 | **0.583** | 0.278 |
| 26 | **0.868** | 0.118 | 0.090 |
| Eigenvalues | **13.76** | **1.75** | **2.43** |

## Reliability

Internal consistency was assessed using Cronbach's alpha (α), with the STP-Q total scale exhibiting an α of 0.954. This indicates an excellent level of consistency within the questionnaire, affirming the interrelatedness of the items. The test-retest reliability, determined by the intraclass correlation coefficient (ICC), was found to be 0.851 (95% CI: 0.801–0.889), signifying a high degree of stability over the assessment interval. The Spearman's rho (ρ) correlation coefficient for the total scale stood at 0.654 (p < 0.001), reflecting a strong positive correlation and reinforcing the test's reliability over repeated administrations (Table 4). Precision in measurement, as evidenced by the standard error of measurement (SEM), was calculated at 2.594, illustrating the scale's accuracy in score estimation. The smallest real difference (SRD95) was established at 7.19, identifying the minimum change considered significant and exceeding the threshold of measurement error for 95% of the sample. Collectively, these reliability indices

**Table 3. Interrelations between STP-Q, E-learning readiness scale, and NMP-Q.**

| | E-Learning Readiness | | NMP-Q | |
|---|---|---|---|---|
| | **r** | **p** | **r** | **p** |
| **STP-Q total score** | 0.110 | 0.034 | 0.336 | p < 0.001 |

STP-Q: Smart Tools Proneness Questionnaire, NMP-Q: Nomophobia Questionnaire.

**Table 4. Reliability statistics for the total scale.**

| | Cronbach's a | Test—retest reliability ICC | Test—retest reliability rho (p) | SEM | SRD95 |
|---|---|---|---|---|---|
| **STP-Q total** | 0.954 | 0.851 (0.801–0.889) | .654 (p<0.001) | 2.594 | 7.19 |

ICC: Intraclass Correlation Coefficient, 95% CI: 95% Confidence Interval, rho (p): Spearman's rank correlation coefficient, SEM: Standard Error of Measurement, SRD95: Smallest Real Difference at 95% Confidence

suggest that the STP-Q is a robust instrument for gauging individuals' propensity to use smart tools, with strong internal consistency and temporal stability as key features of its measurement properties. The Bland-Altman plot, illustrated in Fig 2, provided a visual representation of this agreement, plotting the difference between the test-retest scores against the mean of those scores. (Fig 2. Bland-Altman Analysis Depicting Agreement of STP-Q Test-Retest Measurements) Table 5 presents a detailed examination of the STP-Q scores, focusing on the inter-item consistency and the stability of the scale's internal consistency.

## Discussion

The primary goal of this study is to test the reliability and validity of the Turkish version of the STP-Q. Our research encompasses the adaptation of an instrument that will facilitate a profound understanding of the role smart tools play in our daily lives, ensuring cultural sensitivity and linguistic accuracy for the Turkish community. During the preparation of the Turkish version, the simplicity and comprehensibility of the original English scale were considered to create an exceedingly intelligible structure for the Turkish population. This process followed the standards set by Beaton et al., involving a two-stage translation and subsequent back-translation method [16]. The initial translation, independently performed by two native Turkish speakers, was then back-translated into English by two separate native English speakers, allowing for a meticulous evaluation of semantic integrity and linguistic correctness. This rigorous approach has affirmed the fidelity of the STP-Q's Turkish adaptation to the original text.

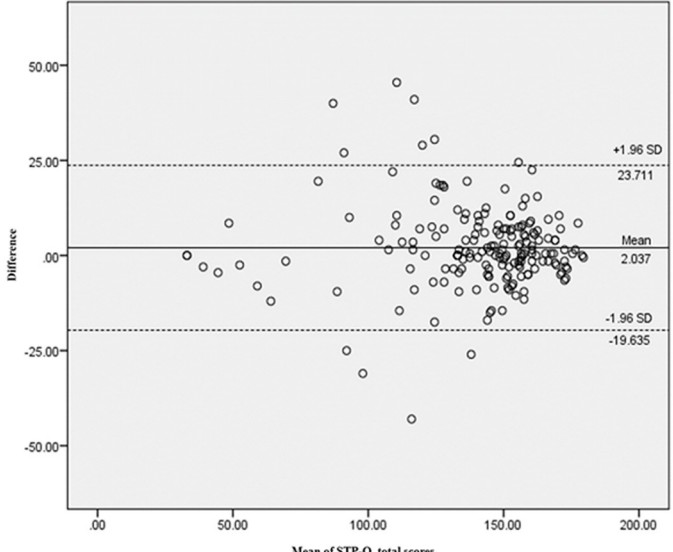

**Fig 2.**

**Table 5. Corrected item-total correlations and Cronbach's a if item deleted results for the STP-Q.**

| Item Corrected | Mean | SD | Corrected item-total correlation | Cronbach's Alpha if Item Deleted |
|---|---|---|---|---|
| 1 | 3.3414 | 1.68055 | .266 | .956 |
| 2 | 4.1663 | 1.93883 | .391 | .955 |
| 3 | 4.0924 | 1.86348 | .407 | .955 |
| 4 | 4.1090 | 1.78707 | .414 | .955 |
| 5 | 4.9918 | 1.77594 | .678 | .952 |
| 6 | 4.9378 | 1.85382 | .684 | .952 |
| 7 | 5.6603 | 1.73455 | .823 | .951 |
| 8 | 5.6999 | 1.69713 | .848 | .951 |
| 9 | 4.9838 | 1.89301 | .633 | .953 |
| 10 | 5.8221 | 1.64420 | .861 | .951 |
| 11 | 5.2108 | 1.83476 | .741 | .952 |
| 12 | 4.4266 | 2.14855 | .443 | .955 |
| 13 | 5.2695 | 1.90946 | .697 | .952 |
| 14 | 5.5473 | 1.82580 | .821 | .951 |
| 15 | 5.2926 | 1.85442 | .703 | .952 |
| 16 | 5.6351 | 1.71617 | .841 | .951 |
| 17 | 4.4204 | 2.05623 | .178 | .958 |
| 18 | 5.5081 | 1.79185 | .771 | .951 |
| 19 | 5.6287 | 1.71017 | .841 | .951 |
| 20 | 5.7385 | 1.76423 | .840 | .951 |
| 21 | 5.2425 | 1.87311 | .768 | .951 |
| 22 | 5.7243 | 1.70680 | .846 | .953 |
| 23 | 5.2405 | 1.90985 | .632 | .952 |
| 24 | 5.5837 | 1.70998 | .763 | .955 |
| 25 | 4.1378 | 2.11556 | .410 | .951 |
| 26 | 5.8544 | 1.70542 | .816 | .953 |
| 27 | 4.9621 | 2.00908 | .601 | .958 |

SD: Standard Deviation, STP-Q: Smart Tools Proneness Questionnaire.

Cross-cultural adaptations underscore the intricate ways in which individual inclinations are not solely the product of personal experiences and abilities but are also significantly shaped by the wider societal and cultural milieu [27]. Tools that are both precise and reliable hold immense value for deciphering the complex global patterns of technology adoption and use [28]. The adaptation of the STP-Q into Turkish offers a comprehensive dataset that sheds light on the cross-cultural and societal nuances influencing technology adoption behaviors.

The demographic research revealed unforeseen patterns. Younger individuals exhibited significantly greater levels of smart tool utilization in comparison to their older counterparts, highlighting the impact of age on the adoption of technology. Surprisingly, there was no significant correlation between BMI and smart tool usage. This indicates that the adoption of smart tools among adults with higher BMI is not primarily driven by the need to monitor their health. Moreover, the increased involvement of female participants in using smart tools for social and hedonic purposes emphasizes the disparities in technology preferences and usage patterns between genders. Literature suggests that women's interactions with technology can be distinct in terms of both behavior and attitude [29]. For example, research indicates that women may utilize smart tools for social connectivity more than men, who often use

technology for information and entertainment purposes [30]. Additionally, the engagement of a predominantly female sample in this study may also be indicative of the growing trend of digital inclusivity and the breaking down of gender barriers in technology use. Studies show that women are rapidly closing the digital divide, not only in terms of usage but also in the fields of technology creation and leadership [31].

The consistency of the average age of participants and expected adult users of smart tools is consistent with the literature; It is stated in the literature that adults are increasingly incorporating smart technology into their daily lives, integrating devices for various functional and hedonic purposes [32, 33]. It is also worthwhile to consider the potential influence of cultural factors on smart tool usage. Cultural values and norms play a crucial role in shaping how individuals interact with technology [34]. Therefore, the demographic distribution within this study may also offer insights into the cultural nuances of smart tool adoption in the Turkish context.

The factor analysis of the Turkish version of the STP-Q has preserved the structural characteristics of the original English version, delineating three distinct factors: (1) utilitarian use, (2) hedonic and social use, and (3) proneness to delegate tasks. Factor 1 was identified as the most significant in our factor analysis based on its elevated eigenvalue and its ability to account for a substantial part of the variation. This component generally encompasses elements associated with the utilitarian utilization of smart tools, exemplifying practical, efficiency-oriented applications of technology in everyday life. The significance of this element emphasizes the pivotal importance that functional and pragmatic features of smart tool usage have for users, suggesting that the main motivation for using smart tools is their capacity to offer practical advantages and enhance efficiency in daily tasks. The results indicated that several items, including items 5, 23, 27, and 17, were removed from the analysis because they had low factor loadings. The exclusion criteria were established by assessing the factor loadings obtained from the exploratory factor analysis (EFA). Items with factor loadings below 0.4 were considered insignificant and were removed since they did not significantly contribute to the underlying constructs examined by the STP-Q. Furthermore, items that demonstrated cross-loading on numerous variables, suggesting a lack of differentiation, were also considered for possible removal. As an illustration, item 5 displayed a factor loading of 0.35, indicating a weak association with its intended component. Item 23, with a factor loading of 0.38, did not provide a substantial contribution to the construct it was designed to measure. Neither item 27 nor item 17, with factor loadings of 0.32 and 0.36 respectively, met the necessary threshold. The elimination of these items was necessary to enhance the overall precision and uniformity of the instrument. To ensure the accuracy and reliability of the constructs assessed by the STP-Q, items with low factor loadings or those that did not clearly match with a single factor were eliminated.

The modest correlation between the STP-Q total score and the E-Learning Readiness Scale suggests that while there is a statistically significant relationship between smart tool propensity and e-learning readiness, the overlap between these constructs is relatively limited. This modest correlation indicates that the STP-Q captures a broader range of behaviors and attitudes toward smart tool usage that are not entirely encompassed by e-learning readiness alone. The distinctiveness of the STP-Q highlights its unique contribution to understanding smart tool usage beyond the scope of e-learning readiness, thereby supporting its convergent validity by demonstrating that it measures related yet distinct constructs. This distinction aligns with literature differentiating general technology inclination from preparedness for specific technological applications [35].

In a more pronounced manner, the STP-Q's correlation with the NMP-Q underlines a more substantial connection, suggesting a meaningful interplay between general smart tool usage and the anxiety related to the absence of these tools. This significant correlation

resonates with current discussions in the literature regarding technology dependence and the psychological implications of constant connectivity [1, 36–38]. Nomophobia, characterized as an increasing concern in our digitally saturated society, serves as a potential indicator of a wider dependency on technology, thus supporting the STP-Q's capacity to reflect multifaceted behavioral tendencies. This evidence of convergent validity reinforces the STP-Q's ability to capture the complex spectrum of our relationship with smart tools, which includes utilitarian purposes, pleasure-driven use, and the anxiety associated with disconnection. The linkage between smart tool affinity and nomophobia may illuminate the intricate emotional and cognitive processes involved in our interactions with digital technologies, offering a broader understanding of technological engagement in contemporary life.

The reliability assessment of the Turkish STP-Q has demonstrated an impressive level of internal consistency, a fundamental feature for the validity of any psychometric instrument. This internal harmony is essential, as it ensures that the various items on the scale collectively reflect the singular construct of smart tool usage. This level of consistency is in line with established norms in the field, as seen in other pivotal studies where high internal reliability is a mark of a well-constructed questionnaire [39–41]. The Turkish STP-Q's temporal stability, highlighted by its test-retest reliability, is a testament to its dependability over time. Such stability is paramount in longitudinal studies or in evaluations pre- and post-interventions designed to detect behavioral shifts. The strong positive correlation observed upon repeated administration of the STP-Q suggests reliable reproducibility of results, bolstering the instrument's credibility for consistent application. Comparative to the original validation studies, the Turkish STP-Q's reliability is affirming. It upholds the rigorous standards set forth in the instrument's inception, demonstrating that the core integrity of the STP-Q remains intact across cultural and linguistic adaptations. As technology becomes increasingly integrated into daily life, understanding these patterns becomes more critical. The STP-Q's ability to measure with such precision offers valuable insights for designing user-centered smart technologies and can inform policies and practices aimed at optimizing digital tool use [42].

A primary limitation of this study is the possibility of sample bias, given that the majority of participants were from urban areas and may not accurately reflect the wider population. Future research should strive to incorporate a broader range of participants, including individuals from various geographical areas and socio-economic statuses, in order to improve the generalizability of the findings. Moreover, the demographic profile, particularly the predominance of female participants, mirrors general societal trends observed in technology use and preferences, which may influence the outcomes.

Moreover, the dynamic and fast-changing nature of technology poses a challenge to the relevance and applicability of the STP-Q. With the emergence of new smart tools and technologies, it is necessary to periodically update the questionnaire to ensure its relevance and validity. Researchers should regularly update the STP-Q to integrate technological advancements and adapt to changes in user behavior.

Additional research should also investigate longitudinal studies to evaluate the evolution of smart tool utilization over an extended period. Such studies have the potential to offer useful insights on the development of individuals' interactions with smart tools and the lasting effects of using smart tools on different psychological and behavioral outcomes. Furthermore, investigating the effectiveness of interventions aimed at enhancing the utilization of smart tools and minimizing any adverse consequences could be a promising field of investigation.

The STP-Q has great potential for applicability in various cultural situations, considering its effective adaptation for the Turkish community. By aligning the questionnaire with the cultural realities of many contexts, the STP-Q can yield precise and significant insights about the inclination towards smart gadgets. This can facilitate the development of culturally sensitive

technologies and interventions that have a higher likelihood of being embraced and successful. Future research should prioritize the adaptation of the STP-Q to different cultural contexts, taking into account linguistic variations and cultural influences that may affect the understanding and significance of its components.

## Conclusion

In conclusion, this study has validated the Turkish adaptation of the Smart Tools Proneness Questionnaire (STP-Q), confirming its reliability and structural integrity. The STP-Q's strong internal consistency and test-retest reliability suggest that it is a robust measure for evaluating the propensity to use smart tools in the Turkish context. The correlations with the E-Learning Readiness Scale and NMP-Q highlight its convergent validity, particularly noting the significant behavioral link to nomophobia. The demographic trends observed in this research add depth to the external validity, ensuring the STP-Q's relevance for contemporary technology engagement. This study paves the way for further research into smart tool use across cultures and contributes to our understanding of digital interaction in Turkiye.

The STP-Q is an invaluable instrument for researchers aiming to delve into the complexities of smart tool usage across diverse demographics. It provides a comprehensive assessment that can be employed to examine the psychological and behavioral impacts of smart tool utilization, facilitating the identification of patterns and trends that can inform future research endeavors. The STP-Q offers educators crucial insights into students' interactions with smart tools, enabling the customization of pedagogical strategies to optimize learning outcomes and effectively integrate technology. By understanding students' propensities for smart tool usage, educators can devise more efficacious instructional methods that leverage technology to enhance educational achievement.

Policymakers can also benefit from the insights provided by the STP-Q. The data obtained through this instrument can inform policy decisions regarding technology adoption and the promotion of digital inclusion. By identifying the barriers and facilitators of smart tool usage, policymakers can design interventions and strategies that ensure equitable access to technology and support its effective utilization across various segments of the population. In conclusion, the STP-Q is a versatile tool with significant applications in research, education, and policy-making. Its comprehensive framework allows for a thorough understanding of smart tool usage, rendering it an essential resource for guiding interventions and strategies aimed at enhancing technology adoption and utilization.

## Acknowledgments

We extend our sincerest gratitude to Dr. Jordan Navarro and colleagues, key developers of the STP-Q, for granting us the permission to adapt the STP-Q into Turkish and for their approval of the final Turkish version. Their contributions were instrumental in the successful localization of this instrument, ensuring its relevance and applicability within the Turkish context.

Furthermore, we acknowledge that none of the researchers involved in this study are native English speakers. To ensure the linguistic quality and fluency of our manuscript, we utilized an artificial intelligence application, Academic Assistant Pro. This tool has been invaluable in refining our language use and enhancing the overall readability of our academic work. Our appreciation extends to all individuals and tools that have supported us in bringing this research to fruition.

## Author Contributions

**Conceptualization:** Ceyhun Türkmen.

**Data curation:** Ceyhun Türkmen.

**Formal analysis:** Serdar Yılmaz Esen, Ceyhun Türkmen.

**Funding acquisition:** Tülin Düger.

**Investigation:** Ceyhun Türkmen.

**Methodology:** Serdar Yılmaz Esen, Ceyhun Türkmen.

**Project administration:** Ceyhun Türkmen, Tülin Düger.

**Resources:** Serdar Yılmaz Esen, Ceyhun Türkmen.

**Software:** Serdar Yılmaz Esen, Tülin Düger.

**Supervision:** Tülin Düger.

**Validation:** Tülin Düger.

**Visualization:** Tülin Düger.

**Writing – original draft:** Serdar Yılmaz Esen, Ceyhun Türkmen.

**Writing – review & editing:** Ceyhun Türkmen, Tülin Düger.

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
