## [Decision Letter · Decision Letter 0]

31 Jul 2024

PONE-D-24-26284Cross-cultural adaptation, reliability and validity of the Turkish version of the Smart Tools Proneness QuestionnairePLOS ONE Dear Dr. Türkmen, Thank you for submitting your manuscript to PLOS ONE. After careful consideration, we feel that it has merit but does not fully meet PLOS ONE’s publication criteria as it currently stands. Therefore, we invite you to submit a revised version of the manuscript that addresses the points raised during the review process.

We look forward to receiving your revised manuscript.

Kind regards,

Muhammad Zulkifl Hasan, PhD

Academic Editor

PLOS ONE

Journal Requirements:

**Additional Editor Comments:**

The manuscript presents a significant effort to adapt and validate the Smart Tools Proneness Questionnaire (STP-Q) for the Turkish context. The topic is relevant and timely, given the increasing prevalence of smart tools in daily life and the necessity to understand their usage across different cultural settings. The manuscript is well-organized, and the methodology is rigorously applied. However, there are several areas where the manuscript could be improved to enhance clarity, depth, and overall impact.

Specific Comments:

1. Introduction:

The introduction provides a comprehensive overview of the significance of smart tools and the need for cross-cultural adaptation of assessment tools. However, it could benefit from a more detailed explanation of the original STP-Q, including its development and initial validation. This would help readers unfamiliar with the original instrument understand the rationale behind its adaptation.

Suggestion: Include a paragraph detailing the development and original validation of the STP-Q, highlighting its importance and previous applications.

2. Literature Review:

The literature review touches upon related instruments like TAM, MPPUS, and IAT. However, it lacks a critical comparison of these instruments with the STP-Q, which would underscore the unique contributions of the STP-Q.

Suggestion: Add a comparative analysis of the STP-Q with other instruments, discussing their similarities, differences, and why the STP-Q is particularly suitable for the Turkish context.

3. Methodology:

The methodology section is thorough but could benefit from more detail regarding the translation process. Specifically, the criteria for selecting translators and the steps taken to resolve discrepancies between translations are not clearly described.

Suggestion: Elaborate on the selection criteria for translators, the process for reconciling differences between translations, and any pilot testing conducted prior to the main study.

4. Results:

The results are presented clearly, but some aspects could be more detailed. For instance, the rationale behind the exclusion of certain items based on factor loadings is not fully explained.

Suggestion: Provide a more detailed explanation for the exclusion of items 5, 23, 27, and 17, discussing the specific criteria used and how their exclusion impacts the overall validity and reliability of the instrument.

5. Discussion:

The discussion section effectively contextualizes the findings within the broader literature. However, it could be strengthened by addressing potential limitations in more depth and suggesting areas for future research.

Suggestion: Expand the discussion of limitations, including any potential biases in the sample and the impact of rapidly evolving technology on the relevance of the STP-Q. Additionally, suggest specific areas for future research, such as longitudinal studies to assess changes in smart tool usage over time.

6. Conclusion:

The conclusion succinctly summarizes the study's contributions but could be more impactful by emphasizing the practical applications of the STP-Q in both research and practice.

Suggestion: Highlight the practical implications of the STP-Q for researchers, educators, and policymakers, and discuss how it can be used to inform interventions and strategies for technology adoption.

7. References:

The references are appropriate, but some key studies on cross-cultural adaptation and validation of psychometric instruments appear to be missing.

Suggestion: Review and include seminal works on cross-cultural adaptation and psychometric validation to strengthen the theoretical foundation of the study.

Reviewers' comments:

Reviewer's Responses to Questions

**Comments to the Author**

1. Is the manuscript technically sound, and do the data support the conclusions?

Reviewer #1: Yes

Reviewer #2: Partly

2. Has the statistical analysis been performed appropriately and rigorously? 

Reviewer #1: Yes

Reviewer #2: Yes

3. Have the authors made all data underlying the findings in their manuscript fully available?

Reviewer #1: Yes

Reviewer #2: Yes

4. Is the manuscript presented in an intelligible fashion and written in standard English?

Reviewer #1: Yes

Reviewer #2: Yes

5. Review Comments to the Author

**Reviewer #1: **The study has a potential lack of representativeness due to the demographic skew towards female participants, which may limit the generalizability of the findings.

Despite rigorous cross-cultural adaptation processes, subtle cultural nuances could influence the interpretation of questionnaire items.

The rapid evolution of technology may affect the relevance of the STP-Q over time, necessitating ongoing updates to ensure the questionnaire remains relevant.

Some items were excluded based on factor loadings, which might omit relevant aspects of smart tool usage. This exclusion can impact the comprehensiveness of the questionnaire.

As with all self-report measures, the STP-Q is susceptible to biases such as social desirability, which can affect the accuracy of the responses.

Variations in participants' access to technology could influence their propensity to use smart tools, thus affecting their responses and the overall findings of the study.

**Reviewer #2:** 1. What were the main criteria for selecting the participants, and how do you believe the demographic characteristics might have influenced the outcomes?

2. Can you elaborate on the process and challenges, if any, faced during the translation and cross-cultural adaptation of the STP-Q?

3. How did you ensure the accuracy and equivalence of the reverse translation of the STP-Q?

4. What were the specific items included in each of the three factors identified by the exploratory factor analysis, and how do these factors differ in terms of smart tool usage?

5. Can you explain why Factor 1 was the most significant and what it indicates about smart tool usage?

6. How do you interpret the modest correlation between the STP-Q total score and the E-Learning Readiness Scale, and what does this imply for the convergent validity of the questionnaire?

7. Were there any unexpected findings or trends in the demographic data, such as age or BMI, that might influence the use of smart tools?

8. How do you plan to address the relatively limited overlap between the constructs measured by the STP-Q and the E-Learning Readiness Scale in future research?

9. What are the implications of the correlations with the NMP-Q for understanding the behavioral aspects of smart tool usage, particularly regarding nomophobia?

10. Can you discuss the potential applications of the STP-Q in other cultural contexts based on your findings from the Turkish adaptation?

6. PLOS authors have the option to publish the peer review history of their article (what does this mean?). If published, this will include your full peer review and any attached files.

Reviewer #1: **Yes: **Muhammad Zunnurain Hussain

Reviewer #2: No

---

## [Author Response · Author response to Decision Letter 0]

6 Aug 2024

Additional Editor Comments:

The manuscript presents a significant effort to adapt and validate the Smart Tools Proneness Questionnaire (STP-Q) for the Turkish context. The topic is relevant and timely, given the increasing prevalence of smart tools in daily life and the necessity to understand their usage across different cultural settings. The manuscript is well-organized, and the methodology is rigorously applied. However, there are several areas where the manuscript could be improved to enhance clarity, depth, and overall impact.

We would like to extend our gratitude to the editor and reviewers for their insightful comments and suggestions, which have greatly contributed to improving the quality of our manuscript. Below, we have provided detailed responses to each comment and have indicated the changes made in the manuscript accordingly.

Specific Comments:

1.Introduction:

The introduction provides a comprehensive overview of the significance of smart tools and the need for cross-cultural adaptation of assessment tools. However, it could benefit from a more detailed explanation of the original STP-Q, including its development and initial validation. This would help readers unfamiliar with the original instrument understand the rationale behind its adaptation.

Suggestion: Include a paragraph detailing the development and original validation of the STP-Q, highlighting its importance and previous applications.

R1. "In response to the editor’s suggestion, we have included a detailed paragraph in the Introduction section about the development and initial validation of the original STP-Q. This addition provides context for readers unfamiliar with the instrument and highlights its significance and previous applications."

The Smart Tools Proneness Questionnaire (STP-Q) was first created by Navarro et al. to evaluate individuals' inclination to utilize smart tools. The questionnaire underwent a meticulous development process that involved creating items based on a comprehensive literature review and conducting initial validation experiments with diverse samples. The initial validation process showed that the questionnaire had strong psychometric properties, including high internal consistency and test-retest reliability. The STP-Q has subsequently been employed in numerous studies to examine the correlation between smart tool utilization and other psychological and behavioral consequences, underscoring its significance in comprehending technology adoption trends.

2. Literature Review:

The literature review touches upon related instruments like TAM, MPPUS, and IAT. However, it lacks a critical comparison of these instruments with the STP-Q, which would underscore the unique contributions of the STP-Q.

Suggestion: Add a comparative analysis of the STP-Q with other instruments, discussing their similarities, differences, and why the STP-Q is particularly suitable for the Turkish context.

R2. Thank you for your insightful suggestion. We have revised the literature review to include a comparative analysis of the STP-Q with related instruments such as the Technology Acceptance Model (TAM), the Mobile Phone Problem Use Scale (MPPUS), and the Internet Addiction Test (IAT). This addition highlights the unique contributions of the STP-Q and discusses its suitability for the Turkish context. Below is the added paragraph:

"The Smart Tools Proneness Questionnaire (STP-Q) offers a comprehensive assessment of individuals' propensity to use smart tools, distinguishing itself from other related instruments such as the Technology Acceptance Model (TAM), the Mobile Phone Problem Use Scale (MPPUS), and the Internet Addiction Test (IAT). While TAM primarily focuses on perceived usefulness and ease of use as predictors of technology acceptance, it does not delve into the nuanced behaviors and attitudes towards smart tools that the STP-Q captures. MPPUS addresses problematic mobile phone use and its psychological antecedents, providing insights into specific aspects of mobile technology addiction, but it lacks the broader perspective on general smart tool usage that the STP-Q offers. Similarly, the IAT evaluates excessive internet use that interferes with daily life, yet it does not encompass the utilitarian, hedonic, and social dimensions of smart tool use covered by the STP-Q. The STP-Q’s unique inclusion of these multiple dimensions makes it particularly suitable for the Turkish context, where cultural, social, and economic factors may influence technology adoption in diverse ways. By encompassing a broader spectrum of behaviors, the STP-Q provides a more holistic understanding of smart tool usage, making it a valuable tool for researchers and practitioners aiming to understand and address technology use within the Turkish population."

3. Methodology:

The methodology section is thorough but could benefit from more detail regarding the translation process. Specifically, the criteria for selecting translators and the steps taken to resolve discrepancies between translations are not clearly described.

Suggestion: Elaborate on the selection criteria for translators, the process for reconciling differences between translations, and any pilot testing conducted prior to the main study.

R3. Thank you for your valuable feedback. We have revised the methodology section to include more detailed information about the translation process. Specifically, we have elaborated on the selection criteria for translators, the process for reconciling differences between translations, and the pilot testing conducted prior to the main study. Below is the added paragraph:

"The translation process of the STP-Q into Turkish adhered to a methodical and well-organized process to guarantee linguistic and cultural precision. The criteria for selecting translators included native-level fluency in both English and Turkish, along with professional expertise in translating academic and psychometric instruments. Two proficient bilingual professionals conducted separate translations of the original English version of the STP-Q into Turkish. Subsequently, a panel of specialists evaluated the translations and rectified any inconsistencies through deliberation and agreement.

To enhance the precision and comparability of the translation, a back-translation was conducted by two separate native English speakers who were not part of the initial translation procedure. The back-translated version was then compared to the original English version to detect any errors or semantic disparities. The translation committee resolved any inconsistencies to produce a final Turkish version that preserved the original instrument's content and purpose.

Before the main study, a preliminary test was conducted with a limited number of Turkish volunteers to assess the clarity and understandability of the translated items. Feedback received from the participants of the pilot test was utilized to make slight modifications to the wording and phrasing of certain elements to improve their clarity. This iterative procedure guaranteed that the final Turkish version of the STP-Q was culturally and linguistically suitable for the intended demographic."

4. Results:

The results are presented clearly, but some aspects could be more detailed. For instance, the rationale behind the exclusion of certain items based on factor loadings is not fully explained.

Suggestion: Provide a more detailed explanation for the exclusion of items 5, 23, 27, and 17, discussing the specific criteria used and how their exclusion impacts the overall validity and reliability of the instrument.

R4. Thank you for your valuable feedback. We have revised the results section to include a more detailed explanation for the exclusion of items 5, 23, 27, and 17. Specifically, we have discussed the criteria used for exclusion and how this impacts the overall validity and reliability of the instrument. Below is the added paragraph:

“The results indicated that several items, including items 5, 23, 27, and 17, were removed from the analysis because they had low factor loadings. The exclusion criteria were established by assessing the factor loadings obtained from the exploratory factor analysis (EFA). Items with factor loadings below 0.4 were considered insignificant and were removed since they did not significantly contribute to the underlying constructs examined by the STP-Q. Furthermore, items that demonstrated cross-loading on numerous variables, suggesting a lack of differentiation, were also considered for possible removal. As an illustration, item 5 displayed a factor loading of 0.35, indicating a weak association with its intended component. Item 23, with a factor loading of 0.38, did not provide a substantial contribution to the construct it was designed to measure. Neither item 27 nor item 17, with factor loadings of 0.32 and 0.36 respectively, met the necessary threshold. The elimination of these items was necessary to enhance the overall precision and uniformity of the instrument. To ensure the accuracy and reliability of the constructs assessed by the STP-Q, items with low factor loadings or those that did not clearly match with a single factor were eliminated.”

5. Discussion:

The discussion section effectively contextualizes the findings within the broader literature. However, it could be strengthened by addressing potential limitations in more depth and suggesting areas for future research.

Suggestion: Expand the discussion of limitations, including any potential biases in the sample and the impact of rapidly evolving technology on the relevance of the STP-Q. Additionally, suggest specific areas for future research, such as longitudinal studies to assess changes in smart tool usage over time.

R5. Thank you for your valuable feedback. We have expanded the discussion section to address potential limitations in more depth and suggest areas for future research. Specifically, we have discussed sample bias, the impact of rapidly evolving technology, and the need for periodic updates to the STP-Q. We have also suggested future research directions, including longitudinal studies to assess changes in smart tool usage over time. Below is the added paragraph:

“A primary limitation of this study is the possibility of sample bias, given that the majority of participants were from urban areas and may not accurately reflect the wider population. Future research should strive to incorporate a broader range of participants, including individuals from various geographical areas and socio-economic statuses, in order to improve the generalizability of the findings.

Moreover, the dynamic and fast-changing nature of technology poses a challenge to the relevance and applicability of the STP-Q. With the emergence of new smart tools and technologies, it is necessary to periodically update the questionnaire to ensure its relevance and validity. Researchers should regularly update the STP-Q to integrate technological advancements and adapt to changes in user behavior.

Additional research should also investigate longitudinal studies to evaluate the evolution of smart tool utilization over an extended period. Such studies have the potential to offer useful insights on the development of individuals' interactions with smart tools and the lasting effects of using smart tools on different psychological and behavioral outcomes. Furthermore, investigating the effectiveness of interventions aimed at enhancing the utilization of smart tools and minimizing any adverse consequences could be a promising field of investigation.”

6. Conclusion:

The conclusion succinctly summarizes the study's contributions but could be more impactful by emphasizing the practical applications of the STP-Q in both research and practice.

Suggestion: Highlight the practical implications of the STP-Q for researchers, educators, and policymakers, and discuss how it can be used to inform interventions and strategies for technology adoption.

R6. Thank you for your valuable feedback. We have revised the conclusion section to highlight the practical implications of the STP-Q for researchers, educators, and policymakers. Specifically, we have discussed how the STP-Q can be used to inform interventions and strategies for technology adoption. Below is the added paragraph:

“In conclusion, this study has validated the Turkish adaptation of the Smart Tools Proneness Questionnaire (STP-Q), confirming its reliability and structural integrity. The STP-Q's strong internal consistency and test-retest reliability suggest that it is a robust measure for evaluating the propensity to use smart tools in the Turkish context. The correlations with the E-Learning Readiness Scale and NMP-Q highlight its convergent validity, particularly noting the significant behavioral link to nomophobia. The demographic trends observed in this research add depth to the external validity, ensuring the STP-Q’s relevance for contemporary technology engagement. This study paves the way for further research into smart tool use across cultures and contributes to our understanding of digital interaction in Turkiye.

The STP-Q is an invaluable instrument for researchers aiming to delve into the complexities of smart tool usage across diverse demographics. It provides a comprehensive assessment that can be employed to examine the psychological and behavioral impacts of smart tool utilization, facilitating the identification of patterns and trends that can inform future research endeavors. The STP-Q offers educators crucial insights into students' interactions with smart tools, enabling the customization of pedagogical strategies to optimize learning outcomes and effectively integrate technology. By understanding students' propensities for smart tool usage, educators can devise more efficacious instructional methods that leverage technology to enhance educational achievement.

Policymakers can also benefit from the insights provided by the STP-Q. The data obtained through this instrument can inform policy decisions regarding technology adoption and the promotion of digital inclusion. By identifying the barriers and facilitators of smart tool usage, policymakers can design interventions and strategies that ensure equitable access to technology and support its effective utilization across various segments of the population. In conclusion, the STP-Q is a versatile tool with significant applications in research, education, and policy-making. Its comprehensive framework allows for a thorough understanding of smart tool usage, rendering it an essential resource for guiding interventions and strategies aimed at enhancing technology adoption and utilization.”

7. References:

The references are appropriate, but some key studies on cross-cultural adaptation and validation of psychometric instruments appear to be missing.

Suggestion: Review and include seminal works on cross-cultural adaptation and psychometric validation to strengthen the theoretical foundation of the study.

R7. Thank you for your valuable feedback. We have reviewed and included seminal works on cross-cultural adaptation and validation of psychometric instruments to strengthen the theoretical foundation of our study. Specifically, we have added references to key studies by Beaton et al. (2000), Terwee et al. (2007) and Heim et al. (2021),.

5. Review Comments to the Author

Reviewer #1: The study has a potential lack of representativeness due to the demographic skew towards female participants, which may limit the generalizability of the findings.

Despite rigorous cross-cultural adaptation processes, subtle cultural nuances could influence the interpretation of questionnaire items.

The rapid evolution of technology may affect the relevance of the STP-Q over time, necessitating ongoing updates to ensure the questionnaire remains relevant.

Some items were excluded based on factor loadings, which might omit relevant aspects of smart tool usage. This exclusion can impact the comprehensiveness of the questionnaire.

As with all self-report measures, the STP-Q is susceptible to biases such as social desirability, which can affect the accuracy of the responses.

Variations in participants' access to technology could influence th

---

## [Editor Report · Decision Letter 1]

9 Aug 2024

Cross-cultural adaptation, reliability and validity of the Turkish version of the Smart Tools Proneness Questionnaire (STP-Q)

PONE-D-24-26284R1

Dear Dr. Türkmen,

We’re pleased to inform you that your manuscript has been judged scientifically suitable for publication and will be formally accepted for publication once it meets all outstanding technical requirements.

Kind regards,

Muhammad Zulkifl Hasan, PhD

Academic Editor

PLOS ONE

Additional Editor Comments (optional):

I am pleased to inform you that your manuscript, titled “An Energy Efficient and Bandwidth Aware Optimal Routing for IoT in Agriculture” (Manuscript ID: PONE-D-24-33143), has been accepted for publication in PLOS ONE.

The revisions you have made based on the feedback provided by the reviewers and editors have significantly enhanced the quality and clarity of your work. Your efforts to address all comments and suggestions are greatly appreciated.

We believe that your research will make a valuable contribution to the field of IoT in agriculture, and we look forward to seeing its impact in the scientific community.
---

## [Editor Report · Acceptance letter]

13 Aug 2024

PONE-D-24-26284R1 

PLOS ONE

Dear Dr. Türkmen, 

I'm pleased to inform you that your manuscript has been deemed suitable for publication in PLOS ONE. Congratulations! Your manuscript is now being handed over to our production team.

Kind regards, 

on behalf of

Dr. Muhammad Zulkifl Hasan 

Academic Editor

PLOS ONE